# Mucormycosis of the Central Nervous System

**DOI:** 10.3390/jof5030059

**Published:** 2019-07-08

**Authors:** Amanda Chikley, Ronen Ben-Ami, Dimitrios P Kontoyiannis

**Affiliations:** 1Infectious Diseases Unit, Tel Aviv Sourasky Medical Center, Tel Aviv 64239, Israel; 2Sackler Faculty of Medicine, Tel Aviv University, Tel Aviv 64239, Israel; 3Department of Infectious Diseases, The University of Texas, M.D. Anderson Cancer Center, Houston, TX 77030, USA

**Keywords:** central nervous system, mucormycosis, Mucorales, zygomycosis

## Abstract

Mucormycosis involves the central nervous system by direct extension from infected paranasal sinuses or hematogenous dissemination from the lungs. Incidence rates of this rare disease seem to be rising, with a shift from the rhino-orbital-cerebral syndrome typical of patients with diabetes mellitus and ketoacidosis, to disseminated disease in patients with hematological malignancies. We present our current understanding of the pathobiology, clinical features, and diagnostic and treatment strategies of cerebral mucormycosis. Despite advances in imaging and the availability of novel drugs, cerebral mucormycosis continues to be associated with high rates of death and disability. Emerging molecular diagnostics, advances in experimental systems and the establishment of large patient registries are key components of ongoing efforts to provide a timely diagnosis and effective treatment to patients with cerebral mucormycosis.

## 1. Introduction

Mucormycosis is the second most frequent invasive mold disease after aspergillosis [1,2,3], with rising incidence reported in some countries [4,5,6,7]. The term mucormycosis refers to invasive disease caused by filamentous fungi belonging to the order Mucorales. Patients at risk are those with poorly controlled diabetes mellitus, immunosuppressed patients such as those undergoing treatment for hematological cancer or recipients of solid organ and hematopoietic stem cell transplantation, and persons who have sustained severe trauma to soft tissues, often with direct inoculation of organic matter. Importantly, the clinical presentation of mucormycosis is shaped by the immune characteristics of the host. Thus, patients with diabetes mellitus usually develop rhino-orbital-cerebral mucormycosis (ROCM), patients with hematological neoplasms tend to develop sino-pulmonary disease, and trauma patients present with necrotizing skin and soft tissue infections [8,9]. In all anatomical locations, relentless tissue invasion and infarction secondary to angioinvasion are the hallmarks of mucormycosis [10,11,12,13].

Involvement of the central nervous system (CNS) represents one of the most severe manifestations of mucormycosis, and often determines the survival and functional outcome of the patient [8,9,14]. 

## 2. Epidemiology

Current estimates of the frequency of central nervous system (CNS) mucormycosis are largely based on published case reports and small series [8], which are prone to publication bias. Population-based surveys provide estimates of the overall incidence of mucormycosis, ranging from 0.43 per 1 million population in Spain [15], to 1.73 per million population in the San Francisco Bay area [16]. Incidence rates in France increased from 0.7 per million in 1997 to 1.2 per million in 2006 [4]. Traditionally, autopsy studies have been the gold standard for assessing the organ involvement of invasive mycoses [5,6]. However, autopsy rates have dropped steadily over the past three decades [5,6]. Moreover, autopsy-based studies on organ involvement in invasive mycoses do not provide information on brain involvement, as autopsies frequently exclude the brain or require separate consent from the family for brain inspection [5,17]. 

The frequency of CNS mucormycosis is highly dependent on the population studied (Table 1) [8,18,19,20]. In areas where diabetes mellitus is the predominant risk factor, for example on the Indian subcontinent, sinus disease predominates and the CNS is the third most common site of infection after the paranasal sinuses and orbit [14]. CNS infection is relatively less frequent in case series dominated by patients with hematologic malignancies [7,21], but cancer has nevertheless become the most significant risk factor for CNS mucormycosis.

### 2.1. Diabetes Mellitus

Patients with diabetes mellitus are at risk for rhino-orbital-cerebral mucormycosis (ROCM), specifically in the context of uncontrolled glycemia and diabetic ketoacidosis (DKA) [8]. Of 337 mucormycosis patients with diabetes mellitus reviewed by Roden et al. [8], 66% had disease of the paranasal sinuses, and most of these (43% of the total) had cerebral extension [8]. Similarly, rhinocerebral disease was strongly associated with an underlying diagnosis of diabetes mellitus (11 of 21 patients, 52%) in a European registry of 230 patients with mucormycosis [18], and in the French RetroZygo cohort (16 of 23 patients, 70%) [9].

Unlike the situation in developed countries, where malignancy has emerged as the main risk factor for mucormycosis (see below), uncontrolled diabetes mellitus continues to be the dominant driver of disease in developing nations [23,24,25]. Moreover, in developing countries mucormycosis is frequently reported as the presenting event of diabetes mellitus [24]. For example, in a report from India, as many as 23% of patients with mucormycosis had previously undiagnosed diabetes mellitus [24].

### 2.2. Malignancy

Malignancy, particularly hematological, has replaced diabetes mellitus as the most frequent risk factor associated with mucormycosis of all kinds in developed countries [8,9,22,26]. This epidemiological trend is likely explained by the increasing number of patients receiving intensive chemotherapy, declining rates of uncontrolled diabetes mellitus [1,4,9,26,27,28,29,30], and possibly other treatment-related factors [31]. Referral bias favoring reports from tertiary-level medical centers caring for hematological malignancy patients over community hospitals may also partially explain this trend. In a review of published reports from the 1940s to 1990s, cancer patients accounted for 17% (154/929) of mucormycosis cases [8]. In contrast, patients with hematological cancer accounted for 50% of cases (50/101) in the more contemporary French RetroZygo cohort (2005–2007) [9]. Marr et al. described a two-fold increase in the incidence of mucormycosis in stem cell transplant recipients from 1985 to 1999 [30]. Remarkably, in a recent multicenter analysis of 46 patients with mucormycosis, 42 (92%) had hematological malignancies [26].

Mucormycosis of the CNS seems to be less common among patients with malignancy and recipients of stem cell transplantation, as compared to those with diabetes mellitus [8]. Only sporadic cases were reported in a series of CNS mycoses in cancer patients published in 1994 [21]. This may be due to the propensity of patients with neoplasms for pulmonary mucormycosis, and the relative rarity of rhino-orbital disease [8]. In the review by Roden et al. [8], CNS involvement was noted in 4% of patients with cancer and 11% of stem cell transplantation recipients. Contemporary data suggests that the frequency of CNS mucormycosis has increased [12,32]. In one review of CNS mycoses, Mucorales accounted for 22% of CNS fungal infections, second only to *Aspergillus* spp. [32]. Pagano et al. identified CNS involvement in 11 of 59 patients (19%) with hematologic malignancy and mucormycosis [19]. In a retrospective study performed at a tertiary cancer center, 35% of culture-positive brain mycoses were attributed to Mucorales [12]. Interestingly, Muggeo et al. [33] retrospectively reviewed pediatric cancer patients with mucormycosis, and found rhino-orbital disease in the majority of cases (14/15; 93%) and CNS involvement in 53% (8/15). These findings suggest age-related differences in the pathobiology of cancer-related mucormycosis, and imply that a high index of suspicion of cerebral extension should be maintained, especially in children.

The Bruton tyrosine kinase (Btk) inhibitor ibrutinib, used in the treatment of mantle cell lymphoma and chronic lymphocytic leukemia, has been associated with invasive fungal diseases with frequent involvement of the CNS. Specifically, intracerebral invasive aspergillosis and, to a lesser extent, cryptococcosis and *Pneumocystis jirovecii* pneumonia were reported [34,35,36,37,38,39]. Few cases of mucormycosis have been described so far in patients receiving ibrutinib [40,41,42]. Of 33 patients with invasive fungal disease (IFD) following treatment with ibrutinib reported by Ghez et al. [37], only one had mucormycosis; the site of involvement was not reported. No cases were reported among 16 patients with ibrutinib-related IFD described by Verughese et al. [43]. The role of Btk inhibitors in antifungal immunity is probably dependent on the interplay among multiple disease- and treatment-related factors [36]. Thus, the contribution of Btk inhibitors to the risk of mucormycosis is presently poorly understood. 

### 2.3. Trauma

Mucormycosis after trauma typically involves soft tissues of the limbs, and less frequently the abdominal or thoracic wall, head and neck [20]. Dissemination to the CNS is rare, even in cases related to head trauma. Reviewing 122 cases of post-traumatic mucormycosis reported in the literature, Lelievre et al. found only one case of dissemination to the brain [20]. There were no cases of CNS infection among 16 patients with post-traumatic mucormycosis in the RetroZygo cohort [20]. Cerebral mucormycosis due to direct inoculation of the brain as a result of penetrating head trauma or neurosurgery is extremely rare [44,45,46]. In one reported case, a diabetic woman sustained blunt injury to the forehead triggering sino-orbital mucormycosis with cavernous sinus invasion [46]. 

### 2.4. Injection Drug Use

Injectable drug use is associated with a unique form of isolated cerebral mucormycosis [8], presumably due to hematogenous spread of sporangiospores present in contaminated drugs and drug injection paraphernalia. Overall, 67% of intravenous drug injectors with mucormycosis have CNS involvement, with the majority of those (97%) having isolated cerebral disease [8]. Kerezoudis et al. found 68 reported cases that met their definition for isolated cerebral mucormycosis [47]. The majority of these patients (82%) had a history of intravenous drug abuse and 20% were infected with HIV.

## 3. Pathogenesis

Significant progress has been made in deciphering the virulence factors of Mucorales, as well as some of the molecular events that mediate endothelial cell invasion and damage [48,49,50,51]. Despite this, fundamental questions about the pathobiology of mucormycosis remain unsolved. For example, why do the same Mucorales species cause rhino-orbital disease in diabetic patients and pulmonary disease in patients with hematological malignancies? And, what are the host and pathogen factors involved in CNS invasion?

The primary route of infection is by inhalation of sporangiospores, the asexual spores of Mucorales, leading to invasive sinopulmonary infection in susceptible individuals. From there, CNS penetration occurs either by hematogenous spread or by direct cranial penetration from the paranasal sinuses. Different host innate defenses have specific roles in containing inhaled sporangiospores and preventing their germination and propagation. Dormant sporangiospores inhaled into the paranasal sinuses and airways are resistant to killing by phagocytes of both immunocompetent and immunocompromised hosts [52]. Germination of these spores is a critical event in the pathogenesis of invasive disease, and inhibition of sporangiospore germination by alveolar macrophages is vital to host defense against mucormycosis. Alveolar macrophages of both diabetic and corticosteroid-treated mice are impaired in their capacity to inhibit sporangiospore germination [52,53]. Thus, viable spores are rapidly cleared from the lungs of nondiabetic mice within two to seven days of intranasal inoculation, whereas heavy seeding of the lungs, blood and brain is detected in diabetic mice [53]. Neutrophils damage Mucorales hyphae, and this ability too is significantly impaired in the presence of hyperglycemia and ketoacidosis [54,55].

Involvement of the CNS occurs most frequently (70%) due to contiguous spread from the paranasal sinuses and orbits [8]. The remaining 30% are divided equally between isolated CNS infection (usually in intravenous drug injectors) and hematogenous spread from distant sites of infection [8]. Mucormycosis of the ethmoid sinus carries a particularly high risk of cavernous sinus thrombosis, because the valveless emissary veins draining this sinus traverse the lamina papyracea and facilitate fungal invasion of periorbital tissue, the orbital apex and the cavernous sinus [56,57]. Maxillary sinus infection often spreads to the hard palate and ethmoid sinuses. Infection of the sphenoid sinus can extend into the cavernous sinus, or invade the carotid artery, and from there embolize to the frontal and parietal lobes. Uncommon manifestations of cranial invasion include sagittal sinus thrombosis and epidural and subdural abscess [58]. Meningitis is rare, but when present, it may manifest as obstructive hydrocephalus due to infiltration of the ventricular lining [59,60].

Autopsy studies show that for all three routes, Mucorales enter the brain by way of the cerebral vascular system [11]. Fungal hyphae grow along the internal elastic lamina and extend into the arterial lumen, which becomes obliterated with intravascular thrombosis and intimal hyperplasia [11,13]. Vascular occlusion leads to cerebral infarction and hemorrhagic necrosis, initially without hyphal invasion of brain tissue [11,12,13]. This necrotic hypoxemic environment likely impairs both host innate immune cells and antifungal drugs from accessing and acting on Mucorales. Hyphal invasion of the necrotic brain parenchyma occurs in advanced CNS mucormycosis and is often a preterminal event [61]. The presence of giant cells and granuloma formation imply a relatively preserved immune response and is associated with better outcomes [12].

Because angioinvasion is a hallmark of both local tissue destruction and hematogenous spread in mucormycosis [10], attention has centered on understanding the interaction of Mucorales with endothelial cells. In vitro, *Rhizopus arrhizus* spores and hyphae adhere to endothelial cells and damage them [50]. Glucose-regulated protein 78 (GRP78) was identified as a putative endothelial cell receptor for Mucorales [49]. GRP78 on endothelial cell surface specifically binds germlings of *R. arrhizus* and other Mucorales, and is required for endocytosis and subsequent endothelial cell damage [49]. Interestingly, both iron and glucose induce GRP78 expression and enhance endothelial cell invasion and damage by Mucorales germlings [49]. In vivo, expression of GRP78 is induced in sinus, lung and brain tissue of DKA mice, whereas anti-GRP78 antibodies protect against fatal *R. arrhizus* infection [49]. These findings suggest that GRP78 mediates CNS invasion by Mucorales, but additional work is required to delineate the interaction of Mucorales with CNS endothelial cells. Spore coat protein homologs (CotH), present in Mucorales but absent in non-pathogenic fungi, were identified as fungal ligands for GRP78 [48].

Specific mechanisms of neuroinvasion by Mucorales have not yet been described. Lessons learned from the neuroinvasive Basidiomycete *Cryptococcus neoformans* underscore the full complexity of microvascular host defense and evasion mechanisms employed by the fungus to enter the brain [62]. For example, *C. neoformans* uses various strategies to traverse the blood–brain barrier, among them a “Trojan horse” mechanism where the fungus hijacks host phagocytes and endocytic signaling pathways and uses them as vehicles [62]. Whether such mechanisms may be operative in CNS mucormycosis remains open to investigation.

Acquisition of iron from the environment represents an important Mucorales virulence factor, but direct evidence for its role in CNS invasion is lacking. Iron acquisition is mediated through the production of high affinity iron permeases and siderophores [51]. In the human host, iron in serum and tissues is bound to carrier proteins, such as ferritin, transferrin and lactoferrin. Thus, normal human serum does not support the growth of Mucorales, whereas profuse fungal growth is observed in the sera of patients with DKA in a manner dependent on lower-than-physiologic pH and high concentrations of unbound iron [63]. Similarly, treatment with deferoxamine, an iron chelator that is utilized by Mucorales as a xenosiderophore, increases the susceptibility of patients treated with hemodialysis to mucormycosis [64].

## 4. Microbiology

*Rhizopus* species, principally *R. arrhizus*, are the most frequent species causing microbiologically-confirmed mucormycosis in the majority of case series [8,65]. Other frequently identified species are *Mucor* spp and *Lichtheimia* (formerly *Absidia*) spp. [65]. Together, the genera *Rhizopus*, *Mucor* and *Lichtheimia* account for 70% to 80% of reported mucormycosis cases [65]. Uncommonly recovered species include *Cunninghamella bertholletiae*, *Rhizomucor* spp., *Syncephalastrum racemosum*, *Saksanea vasiformis*, and *Apophysomyces elegans* [8,9,66]. *Rhizopus* spp. are the most frequent species recovered from patients with all forms of CNS mucormycosis, including ROCM, disseminated mucormycosis and primary CNS infection in people who inject drugs [8]. *C. bertholletiae* has a strong predilection for patients with cancer and neutropenia, causing respiratory infection with early angioinvasion and hematogenous dissemination [66]. Of 21 patients with disseminated *C. bertholletiae* infection, 10 (48%) had CNS disease [66]. Unlike infections caused by the more common genera in the Mucoraceae family (*Rhizopus*, *Mucor*, and *Lichtheimia*) and *C. bertholletiae*, those caused by *Apophysomyces elegans* appear to occur primarily in immunocompetent patients [66]. *Lichtheimia* spp., *Apophysomyces elegans* and *Saksanea vasiformis* are strongly associated with post-traumatic mucormycosis, and have only rarely been implicated in CNS disease [14,20,66,67,68]. Isolated renal mucormycosis, caused by *A. elegans,* is reported almost exclusively from India and China [23,24,66,69], but seeding of the CNS is considered rare [14].

## 5. Clinical Features

Patients with CNS mucormycosis may either present with neurological deficit without clinically evident extracranial disease, as in the case of isolated cerebral mucormycosis, or more commonly with rhino-orbito-cerebral or pulmonary disease that progresses to involve the CNS. These scenarios are discussed below with their respective radiographic findings.

### 5.1. Rhino-Orbito-Cerebral Mucormycosis (ROCM)

Mucormycosis of the sinuses involves the turbinates and paranasal sinuses initially. At this stage, symptoms may be indistinguishable from those of more common causes of sinusitis, and include sinus pain, congestion, mouth or facial pain, hyposmia and anosmia [58,70]. On examination, the nasal mucosa appears initially erythematous, then dark purple and black with the onset of tissue necrosis. The presence of necrotic eschars on the nasal mucosa or hard palate and bloody nasal discharge are sentinel signs that should alert clinicians to the possibility of mucormycosis [70,71]. However, necrotic ulcers are absent in about half the cases [70], underscoring the need for a high index of suspicion in any susceptible patient with new sinus symptoms. The rate of progression from sinusitis to invasive disease may be rapid (i.e., within a few days), but is highly variable. Fever is present in fewer than half the cases [70]. Periorbital swelling, proptosis, and ophthalmoplegia are signs of extension into the orbit. Blurred vision and infra-orbital facial numbness indicate invasion of the optic and infraorbital nerves, respectively. The cavernous sinus is often the first intracranial structure to be involved. Cavernous sinus thrombosis may impair the function of the ocular motor nerves III, IV and VI, the ocular nerve and trigeminal nerve branches V1 and V2 that traverse it [25,70]. Hemiparesis, altered consciousness and focal seizures signal brain invasion and infarction [58]. 

### 5.2. Radiological Findings

Sinus mucormycosis is seen on computed tomography (CT) and magnetic resonance imaging (MRI) as nonspecific nodular mucosal thickening. Inflammatory soft tissue infiltration frequently extends to subcutaneous facial tissue and to the infratemporal and temporal fossae [72]. Non-enhancing mucosal tissue within the involved sinuses and turbinates on contrast MRI (the “black turbinate” sign) may be helpful in differentiating mucormycosis from bacterial sinusitis, where mucosal contrast enhancement is usually detected [72,73,74]. This phenomenon is believed to be due to small vessel occlusion and mucosal ischemia. Radiological evidence of bony destruction in the turbinate, sinus walls, orbital wall, skull base, or hard palate is observed in 40% of ROCM cases [74]; hence, absence of bone destruction is not sufficient to exclude mucormycosis [72]. Extension to the orbit results in thickening and lateral displacement of the medial rectus muscle, preseptal edema, proptosis, and orbital fat infiltration, particularly at the orbital apex. 

Son et al. compared clinical features of patients with ROCM and bacterial orbital cellulitis, and found that mucormycosis presented more frequently with extraocular muscle limitation and sinus mucosal thickening on CT imaging, and less frequently with eyelid swelling [75]. However, none of these features could reliably differentiate the two diseases. Air-fluid levels and complete sinus opacification were present in similar proportions in mucormycosis and bacterial sinusitis [75]. The differential diagnosis of ROCM also includes non-infectious inflammatory diseases, such as thyroid orbitopathy, ocular Sweet’s syndrome, idiopathic orbital inflammatory syndrome, and intraorbital masses, including lymphoma, ocular leukemia, metastases, and lacrimal gland tumors [76]. 

The three most frequent imaging findings in intracranial mucormycosis are cavernous sinus thrombosis, brain infarction and internal carotid artery occlusion [77,78]. Brain imaging shows signs of primarily parenchymal involvement, usually involving the inferior parts of the frontal lobes (Figure 1). Lesions may be either hypo- or hyperintense on T2-weighted series [73], and diffusion weighted imaging (DWI) series show markedly reduced diffusion [78]. An intriguing case report outlined the use of magnetic resonance spectroscopy to differentiate CNS mucormycosis from bacterial cerebritis [79]; additional validation of this modality is warranted. Metastatic cancer and the Tolosa–Hunt syndrome are the main differential considerations in patients with cavernous sinus syndrome [25].

### 5.3. Pulmonary Mucormycosis

Pulmonary mucormycosis typically presents in patients with hematologic malignancies and prolonged profound neutropenia as persistent or recurrent fever, cough, and dyspnea, with or without concomitant sinusitis [58,80]. Pleuritic chest pain or the presence of a friction rub may be important clues to infection with an angioinvasive mold [71]. The appearance of focal neurological deficits in such patients is indicative of CNS involvement, either from occult sinus disease or due to hematogenous dissemination from the lungs [81]. The combination of pulmonary infiltrates and CNS lesions in an immunosuppressed patient warrants consideration of a number of opportunistic infections, including invasive aspergillosis, nocardiosis, toxoplasmosis, cryptococcosis, tuberculosis and nontuberculous mycobacteria, as well as metastatic malignancy. In addition, bacterial sepsis with pulmonary septic emboli from intravenous drug injection, central venous catheter infection or right-sided endocarditis, may lead to the development of pyogenic brain abscess [82].

Early differentiation between invasive pulmonary aspergillosis and pulmonary mucormycosis, before dissemination to the CNS, is a clinical challenge that has important therapeutic implications. Chamilos et al. reviewed contemporaneous patients with cancer and pulmonary aspergillosis or mucormycosis, and found that concomitant sinusitis, receipt of voriconazole as antifungal prophylaxis, and the presence on chest CT of ≥10 pulmonary nodules or pleural effusion were significantly associated with a diagnosis of mucormycosis [83]. The presence on chest CT of a ground glass infiltrate surrounded by a dense ring (reversed halo sign), often in the upper lobes, is frequently present early in the course of pulmonary mucormycosis in neutropenic patients [84,85,86], and is unusual in other pulmonary mycoses [86]. CT-guided percutaneous biopsy of pulmonary nodules, coupled with Calcofluor white staining, provides rapid differentiation of non-septate hyphae of mucormycosis from the more frequent septate hyphae of *Aspergillus* spp. [87]. The use of this procedure is limited, however, by the frequent risk of hemorrhage due to thrombocytopenia in patients with hematologic cancers.

### 5.4. Isolated Cerebral Mucormycosis

The typical presentation of this syndrome is the recent onset of altered mental status, headache, hemiplegia and dysarthria in a person with a history of intravenous drug injection [47]. Fever is present in 50% of cases [47]. Brain CT typically shows unilateral basal ganglia hypodensity [88]. Patients are often initially assigned a diagnosis of drug intoxication or ischemic stroke. However, progressive neurological and cognitive deterioration should prompt further investigation. MRI most often shows unilateral basal ganglia mass lesions, with diffusion restriction, varying degrees of contrast enhancement, hemorrhage and perilesional edema [47,88]. Infection may progress rapidly to involve the contralateral basal ganglia [47,88]. Lesions may also appear in the cerebellum and fourth ventricle.

### 5.5. Intracranial Granulomatous Mucormycosis

Intracranial fungal granuloma is a distinct clinical entity, with most cases to date reported from India [89,90]. About half the cases are associated with fungal sinusitis, and half appear as isolated intracranial infections with no clinically apparent sinus disease. *Aspergillus* spp. are the most frequently recovered organisms, followed by Mucorales [89,90]. CT and MRI imaging shows tumor-like masses with faint contrast enhancement and surrounding parenchymal edema [89]. The frontal lobes are the most frequent location [89,90]. Diabetes mellitus is the most frequent predisposing factor, but surprisingly, half of the patients have no obvious risk factors or immune deficiency [89]. Patients present with headache, vomiting, proptosis, and symptoms related to associated sinusitis. Focal neurological deficits are present in a third of patients at presentation [89,90].

## 6. Diagnosis

There are currently no clinically available circulating biomarkers of mucormycosis [91]. Therefore, definitive diagnosis relies on microbiological analyses of tissue obtained by biopsy or surgical debridement [92]. Presumptive diagnosis can be made by direct microscopic examination of fresh tissue. The sensitivity of direct microscopy is greatly enhanced with the use of optical brighteners, such as Calcofluor white, and fluorescence microscopy [92]. Observation on direct microscopy of non-septate (or pauci-septate), broad (6-16 micrometer) hyphae with branching at right angles suggests infection with Mucorales rather than *Aspergillus* species. The non-septate hyphae of Mucorales are susceptible to damage from shear stress [92], and thus tissue grinding may significantly lower the yield of culture. Therefore, homogenization of tissue specimens should be avoided when mucormycosis is suspected. Detection of Mucorales on histopathological examination is enhanced by stains that highlight hyphae, such as Grocott methenamine silver (GMS) and periodic acid Schiff (PAS). Mucorales species may require longer staining times than other fungi [92].

Because a brain biopsy is highly invasive and may lead to neurological deficits, the diagnosis of CNS mucormycosis is frequently made indirectly by identifying the pathogen in the sinuses or lungs. Patients with suspected ROCM should be examined by an Ear, Nose and Throat (ENT) surgeon experienced in the diagnosis and treatment of this disease. Fiberoptic examination of the nasal cavity and septum may reveal ischemic or necrotic mucosal lesions, which should be biopsied. Flexible bronchoscopy and bronchoalveolar lavage (BAL) or CT-guided lung biopsy are used to diagnose pulmonary mucormycosis [87].

Several culture-independent technologies for the diagnosis of mucormycosis from tissue specimens or blood are being developed. A monoclonal antibody (2DA6) against alpha 1,6 linked mannose, which is conserved among the Zygomycota and Ascomycota phyla, appears promising for the detection of a broad range of pathogenic fungi, including Mucorales [93]. 2DA6 was used to develop a lateral flow immunoassay which can be used as a rapid point-of-care test on serum or BAL fluid. PCR-based assays, using either panfungal primers [94] or primers directed at Mucorales specific targets, such as CotH [95], appear promising. Finally, immunohistochemical staining of tissue specimens has shown excellent sensitivity and specificity for Mucorales spp, enhancing morphological differentiation between mucormycosis and aspergillosis [96].

## 7. Treatment

Successful treatment of mucormycosis requires surgical debridement of infected tissue, prompt institution of effective antifungal treatment, and correction of the underlying metabolic and immune derangement (Table 2). Timeliness of these interventions is critical, and nowhere more so than for cerebral mucormycosis. This is because infection may often progress indolently in its initial stages. Early diagnosis may facilitate surgical debridement of involved sino-orbital tissue before cranial invasion occurs.

### 7.1. Surgery

Surgical debridement is considered a mainstay of mucormycosis treatment, and was associated with improved survival in several case series [8,9,18]. In patients with ROCM, sinus surgery is regarded as essential, but the extent of surgical debridement varies from limited to radical resection. Endoscopic surgery has been advocated for early localized invasive fungal sinusitis, whereas open surgery is reserved for patients with orbital and intracranial extension [97,98]. Extensive surgery, such as orbital exenteration and craniofacial resection, is associated with significant morbidity and disfigurement. Importantly, recent studies found no evidence that such radical surgical procedures improve survival rates [99,100,101]. Thus, the extent of surgery warrants careful consideration, taking into account the patient’s underlying comorbidities and life expectancy.

Debridement of infected brain tissue is associated with severe morbidity and uncertain benefits. In a report of the Italian Epidemiological Surveillance of Infections in Hematological Diseases (SEIFEM) group, surgical resection was undertaken less frequently for patients with hematological malignancy and CNS Invasive Fungal Disease (IFD) (10%) than for those with IFD of other sites (33%) [32]. This is likely explained by the purported higher surgical risk of patients with CNS IFD. Moreover, patient selection for surgery may be critical as no significant benefit from surgery was observed in terms of overall survival among unselected patients with CNS IFD [32]. In contrast, stereotactic brain procedures appear safe, even in thrombocytopenic patients, and this may be an underutilized approach [102].

Decisions regarding the surgical approach, its goals and extent require careful discussion among neurosurgeons, infectious diseases specialists, radiologists, patients and their families [103]. Accepted indications for neurosurgery include relief of intracranial pressure, drainage of obstructive hydrocephalus, and excision of lesions compressing the spinal cord. Radical excision of fungal brain abscess or granuloma should be avoided [56]. In cases of hemispheric stroke where elevated intracranial pressure and impending herniation are concerns, decompressive hemicraniectomy may be performed [56]. 

### 7.2. Antifungal Drugs

There are no prospective controlled clinical trials on antifungal drugs for the treatment CNS mucormycosis. In general, antifungals used to treat CNS mycoses must meet several requirements: 1. In vitro, preferably cidal activity against the infecting fungal strain; 2. Adequate penetration of the blood–brain barrier resulting in tissue drug concentrations that reliably exceed the MIC; 3. A favorable safety profile, allowing treatment for extended time periods. Unfortunately, no single drug for mucormycosis meets all of the above requirements. Initial treatment with liposomal amphotericin B (l-AmB) is based on clinical experience and in vitro activity of this agent against the majority of Mucorales isolates. However, amphotericin B achieves low brain tissue concentrations and is associated with nephrotoxicity at high doses. Combining l-AmB with an extended-spectrum azole, either posaconazole or isavuconazole, may increase the likelihood of achieving effective drug concentrations in the brain, but supporting clinical data for such combinations is lacking. Specifically, retrospective analyses of monotherapy versus combination therapy for mucormycosis in patients with hematological malignancies showed no difference in outcomes between treatment strategies [104,105]. Of note, CNS infection was not assessed in these studies. Patients who respond to initial therapy with l-AmB may be switched to maintenance treatment with the less toxic azoles, posaconazole or isavuconazole. The optimal duration of treatment for CNS mucormycosis is unknown. Factors that should be weighed when deciding on the duration of treatment are the extent of surgical debridement performed and the immune state of the patient [106]. Most experts treat CNS mucormycosis for at least six months. For patients with ongoing immunosuppression, such as those with hematological malignancies and recipients of stem cell transplantation, treatment is continued at least for the duration of active immunosuppression. 

Amphotericin B is considered the treatment of choice for mucormycosis, although never assessed in a prospective controlled clinical trial. The liposomal formulation is preferred, as the high doses of amphotericin B deoxycholate (AmBd) required to achieve a fungicidal effect against *Mucorales* spp. are associated with significant toxicity. In mice with experimental mucormycosis, l-AmB (15 mg/kg/day) improved survival whereas amphotericin B deoxycholate (1 mg/kg/day) had no significant effect on morality [107]. Amphotericin B lipid complex (ABLC) achieves higher pulmonary concentrations and clears *R. oryzae* from the lungs more effectively than does l-AmB [108]. Conversely, retrospective clinical data suggest that ABLC may be associated with higher rates of clinical failure in the treatment of ROCM as compared with other amphotericin B formulations [109]. 

Clinical data is mostly retrospective. In an analysis of patients with mucormycosis and hematologic malignancies, delayed initiation of treatment with amphotericin B (≥6 days after diagnosis) was independently associated with increased mortality [110]. The largest prospective therapeutic trial on mucormycosis was a single arm study of high dose L-AmB (10 mg/kg/d) that recruited 40 patients, nine of whom had ROCM [111]. Seventy-one percent of patients underwent surgical debridement. The primary outcome of response at week four occurred in 36% of patients, similar to the 40% global response observed in the DEFEAT study, where the protocol defined l-AmB dose was ≥5 mg/kg thrice weekly (see below) [112]. Overall mortality was 38% at week 12, as compared to 42% in the DEFEAT study. and doubling of the serum creatinine occurred in 40% of patients. These results, though uncontrolled, suggest that high doses of l-AmB are not more effective than standard dosing (≥5 mg/kg/d), and are associated with high rates of nephrotoxicity. Similar findings were noted in an analyses of treatment outcomes in two patient registries [105].

Amphotericin B crosses the blood–brain barrier poorly, and is detectable at low concentrations in the brain parenchyma of patients treated with either l-AmB or AmBd [113,114]. Cerebral Spinal Fluid (CSF)/plasma and brain/plasma concentration ratios of <0.3 have been achieved in rabbits treated with either formulation [115]. Although experimental studies have shown that, unlike the azoles, the penetration of amphotericin B (in any formulation) is not enhanced by meningeal inflammation [115], this drug has been used successfully to treat patients with cryptococcal and Coccidiodes meningitis [116,117]. Intrathecal administration of amphotericin B, either via an external ventricular drain or using an Ommaya reservoir, has been reported in a few cases [118]. The limited data do not allow conclusions to be drawn about the efficacy of this procedure. Previous experience with the use of intrathecal amphotericin B is derived from the treatment of Coccidioides meningitis, where it was associated with significant direct toxicity and was subsequently replaced by safer treatment options [119].

There is limited information on the CNS penetration of isavuconazole, but emerging animal and human data indicate that it is comparable to that of voriconazole. The brain/plasma concentration ratio was 1.8 after a single dose, and close to 1 after repeated dosing [120,121]. These pharmacokinetic features allow therapeutic drug concentrations to be reached in brain parenchyma using standard isavuconazole dosing. However, the drug concentration in the necrotic center of a fungal brain abscess was almost zero [120]. Isavuconazole was assessed prospectively in a single-arm study of patients with mucormycosis and other rare fungal infections (the VITAL trial) [122]. Thirty-seven patients were enrolled, six with CNS mucormycosis. Four patients (11%) had partial response and 16 (43%) had stable disease on study day 42 [122]. The day 42 crude all-cause mortality was 33%, similar to matched historical controls who received amphotericin B-based treatment regimens [122].

CSF concentrations of posaconazole are highly variable in treated patients, with CSF to plasma ratios ranging from 2.3 to <0.01 [123,124,125]. CSF penetration appears to be significantly enhanced in the presence of meningeal inflammation [124]. Posaconazole as an oral suspension was associated with successful treatment in 50% of CNS fungal infections, including cryptococcosis and invasive mold infections [126]. Anecdotal reports describe clinical success in the treatment of cerebral mucormycosis with posaconazole, followed by long-term suppressive treatment with the drug [127,128]. 

Data on the use of in vitro susceptibility testing results (MIC) to guide treatment decisions are very limited. If the infecting strain is grown in culture, antifungal susceptibility testing may be performed using standardized broth microdilution methods. Susceptibility breakpoints have not been established for *Mucorales* spp. 

## 8. Adjunctive Therapeutic Modalities

### 8.1. Iron Chelation

Given the critical role played by iron acquisition in the virulence of Mucorales spp, strategies to limit the availability of iron have been explored as adjunctive therapy in the treatment of mucormycosis. Deferasirox, an iron chelator that is not utilized by Mucorales as a xenosiderophore, was found to have cidal activity against *Mucorales* spp. in vitro, and synergized with l-AmB to improve the survival of mice with experimental mucormycosis [129,130]. A reported case highlighted the potential use of deferasirox as salvage treatment of patients with cerebral mucormycosis [131]. The patient had progressive lesions in his cerebellum and pons after orbital exenteration and treatment with l-AmB. Treatment with deferasirox resulted in rapid clinical and radiological resolution of cerebellar disease [131]. Treatment with l-AmB and deferasirox was compared with l-AmB and placebo in a multicenter, randomized blinded clinical trial (the DEFEAT Mucor study) [112]. Twenty patients were recruited, most with rhino-orbital or pulmonary disease; none had CNS mucormycosis. Death occurred more frequently in the deferasirox arm (30-day mortality, 45% versus 11%, *p* = 0.1; 90-day mortality, 82% versus 22%, *p* = 0.01) [112]. Imbalances were noted between the treatment arms: Active malignancy, neutropenia and corticosteroid treatment, scenarios where iron chelation may be less effective than in DKA, were more common in the deferasirox arm. Nevertheless, until additional clinical studies are performed, deferasirox cannot be recommended for the treatment of mucormycosis [132].

### 8.2. Hyperbaric Oxygen (HBO)

Hyperbaric oxygen (100% oxygen; >1 atmosphere absolute) has transient fungistatic activity in vitro [133,134], and may theoretically reverse some of the tissue hypoxia and acidosis associated with mucormycosis and enhance the antifungal activity of neutrophils, macrophages and amphotericin B. In one report, two patients with ROCM and brain abscess after surgical debridement and AmBd treatment responded to HBO, and remained free of disease 21 months after discharge [135]. In a retrospective analysis of patients with ROCM treated with AmBd and surgical debridement, two of six patients who received adjunctive HBO died versus four of seven who were not treated with HBO [136]. A literature review identified 28 patients with mucormycosis (21 with ROCM) who were treated with HBO [137]. The survival rate was remarkably high (86%), and even higher (94%) when considering only diabetic patients [137]. However, such reports are very likely subject to selection and publication bias. In a retrospective study of 106 patients with hematological malignancies and mucormycosis, survivors were more likely than non-survivors to have received HBO treatment (16% versus 2%, *p* = 0.02). However, HBO was not a significant predictor of survival in the multivariate analysis [104]. In sum, adjunctive HBO is marginally supported by uncontrolled clinical data. Its use can be considered as part of a multimodal treatment approach, specifically in non-neutropenic patients with ROCM. 

### 8.3. Echinocandins

While lacking intrinsic activity against Mucorales, echinocandins have synergistic activity with AmBd and L-AmB against Mucorales, both in vitro and in experimental murine mucormycosis [138]. In a retrospective analysis of 41 patients with ROCM, most of whom had diabetes mellitus, 6 patients treated with caspofungin and amphotericin B had better clinical success and overall survival as compared with patients treated with amphotericin B monotherapy (100% versus 45%, *p* = 0.02) [109]. These findings have not been confirmed in controlled clinical trials. With reference to CNS mucormycosis, echinocandins are large lipopeptides that reach negligible concentrations in the brain and CSF [139]. Given the paucity of supporting clinical data, polyene-echinocandin combination therapy cannot be recommended at present.

## 9. Future Directions

Several unmet needs remain for the treatment of mucormycosis in general and for CNS mucormycosis in particular (Table 3). Despite the availability of improved diagnostics and the introduction of novel antifungals, cerebral mucormycosis remains a highly lethal and disabling disease. As early diagnosis is crucial, efforts to develop and clinically validate circulating biomarkers of mucormycosis are a priority. The antifungal drug pipeline contains molecules that are active against Mucorales, such as APX001 (Amplyx pharmaceuticals) and oteseconazole (formerly VT-1161, Mycovia pharmaceuticals) [140]. Developing a pathophysiologically relevant animal model of cerebral mucormycosis is an important prerequisite for assessing current and emerging treatment options. Finally, performing clinical trials on mucormycosis is extremely challenging, as reflected by the fact that only one prospective controlled trial has been performed to date [112]. The overall rarity of this disease, and its often-rapid progression and severity impede patient recruitment, while the heterogeneity of clinical presentation, underlying comorbidities and treatments complicate data interpretation. Multicenter patient registries may serve as a reasonable alternative to such trials, allowing “real world” data to be collected in order to address important clinical questions, such as outcomes associated with different antifungal treatment strategies and the role, timing and extent of surgery [105].

## Figures and Tables

**Figure 1 jof-05-00059-f001:**
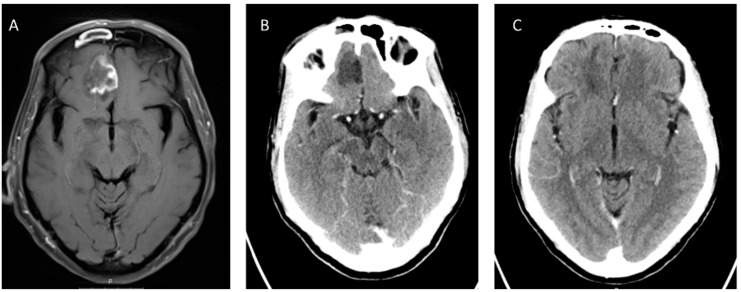
Cerebral mucormycosis. A 67-year old male with acute B-cell lymphoblastic leukemia, chemotherapy-induced neutropenia and high-dose corticosteroid treatment, developed sinopulmonary mucormycosis with cerebral extension involving the right frontal lobe (T1 weighted magnetic resonance (MR) image (**A**) and computed tomography (CT) (**B**)). Culture from sinus biopsy and stereotactic brain biopsy grew *Rhizopus arrhizus*. The patient was treated with Liposomal Amphotericin B (10 mg/kg per day) and Isavuconazole, without debridement of brain tissue. Follow-up CT after 4 months of treatment shows near-complete resolution (**C**). Courtesy: M. Weinberger and N. Carmi Oren.

**Table 1 jof-05-00059-t001:** Patterns of Central Nervous System (CNS) mucormycosis in different patient populations.

Underlying Condition	Proportion of CNS Involvement	Form of CNS Involvement			Reference
	Total	Rhinocerebral	Hematogenous	Isolated CNS	
Diabetes mellitus	43%	43–52%	0%	0%	[8,18]
Malignancy	4–19%	4–15%	12%	0%	[8,18,19]
Stem cell transplantation	11%			0%	[8]
Trauma	1%>		1%>		[20]
Injection drug use	67%	5%		62%	[8]
Overall	12.8–44.1%	11.3%	7.8%	2%	[8,9,14,18,22]

**Table 2 jof-05-00059-t002:** Recommendations for treatment of central nervous system mucormycosis.

	Recommendation
**Surgical treatment**	**Debridement of extracranial site of infection:** Sinus debridement using endoscopic approach for early disease and open surgery for extensive disease.
	**Consider indications for neurosurgery:** Increased intracranial pressure (e.g. hemispheric stroke) Obstructive hydrocephalus Lesions compressing the spinal cord
**Antifungal treatment**	**Initial treatment:** Liposomal amphotericin B 5-10 mg/kg/day IV for initial 28 days. **Alternative:** Isavuconazole 300mg TID for 2 days followed by 300mg QD, IV or PO. **Step-down:** Isavuconazole 300mg TID for 2 days followed by 300mg QD PO.
	**Duration of treatment:** at least 6 months. Factors affecting treatment duration are the extent of surgery done and immune status of the patient.
**Ancillary treatment**	Correction of hyperglycemia and ketoacidosis.
	Discontinue or reduce dose of immunosuppressive drugs, when possible.
	Consider hyperbaric oxygen for rhino-orbito-cerebral mucormycosis.

**Table 3 jof-05-00059-t003:** Unmet needs in the study and management of CNS mucormycosis.

Development of a reproducible and relevant animal model of CNS mucormycosis with validated endpoints of measurement of outcome (e.g., PCR, antigen, and/or histopathology) to offer insights on the pathogenesis and appropriate management of this infection.
Investigate if there is Mucorales species-specific or isolate- specific CNS tropism.
Study the role of brain immune effector cell activity against Mucorales.
Study strategies employed by Mucorales to access the CNS (e.g., through GRP78 attachment, hijacking of host phagocytes and endocytosis).
Development of validated non-culture-based biomarkers in blood and CSF (e.g., based on antigen detection, PCR, volatile organic compounds derived from Mucorales metabolism).
Development of validated neuroimaging readouts that differentiate CNS mucormycosis from other fungal and non-fungal CNS diseases.
Development of rapidly cidal agents that penetrate the blood–brain barrier.
Effect of metabolic derangements (glycemia, ketoacidosis and iron overload) on the degree and rate of CNS involvement.
Risk stratification models to examine the benefit of surgery (type, timing).
Immuno-adjunctive strategies that enhance Mucorales killing in the brain microenvironment without resulting in excess inflammation.
Robust registries of CNS mucormycosis cases.

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
