# Peer review of "Mucormycosis of the Central Nervous System"

_jof, 2019, doi:10.3390/jof5030059_

Round 1

Reviewer 1 Report

This is a terrific review of the state of the field. The authors are amongst the foremost authorities in the world about these challenging infections. The review is excellent in its current form. Except  for minor formatting issues is ready for publication now. A couple of minor additions (if there is enough space to allow this) might be:

1.      A bit of a wider discussion on DM associated cerebral mucormycosis in the developing world and in patients in the developed world with poor access to healthcare

2.     A table summarizing recommendations for treatment

Author Response

Reviewer 1

This is a terrific review of the state of the field. The authors are amongst the foremost authorities in the world about these challenging infections. The review is excellent in its current form. Except  for minor formatting issues is ready for publication now. A couple of minor additions (if there is enough space to allow this) might be:

1.      A bit of a wider discussion on DM associated cerebral mucormycosis in the developing world and in patients in the developed world with poor access to healthcare

2.     A table summarizing recommendations for treatment

Response:

We thank the reviewer for their kind comments.

In response, we have expanded the discussion of diabetes mellitus as a cause of mucormycosis in developing countries [page 4 paragraph 4], noted reports of isolated renal mucormycosis in India and China [page 11 paragraph 1], and added a new table (Table 2 in the revised manuscript) summarizing treatment recommendations.

Reviewer 2 Report

In this review the authors undertake a comprehensive review of CNS mucormycosis. The work is thorough and detailed and I don’t have any major criticisms to give. Addressing some minor points, mostly structural, would improve the manuscript.

In general it is difficult to cover CNS mucormycosis narrowly without providing a good deal of information on this class of infections in general. Overall the authors do a good job of keeping an appropriate balance. In some cases it becomes hard to track whether it is mucormycosis in general or specifically CNS disease that is under discussion:

The introductory paragraph is one such case—it reads like an introduction to the broader topic, only specifically turning to CNS infections at the end. Placing the subtopic under discussion in clear relationship to mucormycosis in general, at this early point in the piece, will help to direct the reader’s interpretation.

The Clinical Features section has a similar problem. Starting with ROCM makes good logical sense but it becomes confusing in the Pulmonary mucormycosis subsection. I believe the intent is to discuss the presentation of CNS involvement in the context of pulmonary mucor, and this expectation should be set more plainly. Part of that issue is having a Radiological findings subsection, which actually relates specifically to ROCM imaging. I became confused about the focus in this section.

     Lastly, on line 70, it would be clearer to say “…most frequent risk factor associated with mucormycosis of all kinds in developed countries”

Smaller points:

The final sentence of the introduction (lines 41-42) seems out of place and doesn’t contribute to the point of the manuscript.

On line 69, the phrase “hematological cancer” is redundant as Malignancy is already stated.

Line 75: “50% of all cases (10/101)” must be a typo.

Concerning the point made on line 77-78: as opposed to a true general shift in epidemiology, the different patient populations seen at tertiary versus general hospitals, combined with the much greater incentives for publishing at the former, could also explain this surprising statistic.

Lines 178-180. While there is certainly evidence for a “Trojan Horse” mechanism in cryptococcal CNS invasion, there is definitely not any consensus that this is the only mechanism, as this sentence seems to imply.

Line 311: “fluorescent microscopy” should be “fluorescence microscopy”

Line 342: “Surgery should be a subsection, as should “Antifungal Drugs” and “Adjunctive therapeutic modalities.”

Lines 420-422: please provide a reference or references.

Line 428: “null” should be “nil.”

Author Response

Reviewer 2

In this review the authors undertake a comprehensive review of CNS mucormycosis. The work is thorough and detailed and I don’t have any major criticisms to give. Addressing some minor points, mostly structural, would improve the manuscript.

In general it is difficult to cover CNS mucormycosis narrowly without providing a good deal of information on this class of infections in general. Overall the authors do a good job of keeping an appropriate balance. In some cases it becomes hard to track whether it is mucormycosis in general or specifically CNS disease that is under discussion:

The introductory paragraph is one such case—it reads like an introduction to the broader topic, only specifically turning to CNS infections at the end. Placing the subtopic under discussion in clear relationship to mucormycosis in general, at this early point in the piece, will help to direct the reader’s interpretation.

The Clinical Features section has a similar problem. Starting with ROCM makes good logical sense but it becomes confusing in the Pulmonary mucormycosis subsection. I believe the intent is to discuss the presentation of CNS involvement in the context of pulmonary mucor, and this expectation should be set more plainly. Part of that issue is having a Radiological findings subsection, which actually relates specifically to ROCM imaging. I became confused about the focus in this section.

     Lastly, on line 70, it would be clearer to say “…most frequent risk factor associated with mucormycosis of all kinds in developed countries”

Response:

We thank the reviewer for their kind comments. We have edited the text to clarify the distinction between mucormycosis in general and CNS mucormycosis. The introduction was broken into 2 sections, the first related to mucormycosis and the second to CNS mucormycosis. We added an introductory paragraph to the “Clinical Features” section, explaining that the following sections describe the evolution of extracranial mucormycosis syndromes (i.e. ROCM and pulmonary mucormycosis) leading to involvement of the CNS. We have edited line 70 as suggested by the reviewer.

Smaller points:

The final sentence of the introduction (lines 41-42) seems out of place and doesn’t contribute to the point of the manuscript.

Response: we agree and have removed this sentence.

On line 69, the phrase “hematological cancer” is redundant as Malignancy is already stated.

Response: this sentence was edited and “cancer” removed.

Line 75: “50% of all cases (10/101)” must be a typo.

Response: Indeed a typo. The figure was corrected to 50/101.

Concerning the point made on line 77-78: as opposed to a true general shift in epidemiology, the different patient populations seen at tertiary versus general hospitals, combined with the much greater incentives for publishing at the former, could also explain this surprising statistic.

Response:

We agree with this point, although we believe referral bias alone cannot sufficiently explain the epidemiological shift in mucormycosis.

We added the following statement to the section:

“Referral bias favoring reports from tertiary-level medical centers caring for hematological malignancy patients over community hospitals may also partially explain this trend.”

Lines 178-180. While there is certainly evidence for a “Trojan Horse” mechanism in cryptococcal CNS invasion, there is definitely not any consensus that this is the only mechanism, as this sentence seems to imply.

Response:

The sentence has been changed as follows:

C. neoformans uses various strategies to traverse the blood-brain barrier, among them a “Trojan horse” mechanism where the fungus hijacks host phagocytes and endocytic signaling pathways and uses them as vehicles”

Line 311: “fluorescent microscopy” should be “fluorescence microscopy”

Response: text was corrected.

Line 342: “Surgery should be a subsection, as should “Antifungal Drugs” and “Adjunctive therapeutic modalities.”

Response: surgery, antifungal drugs and adjunctive therapeutic modalities were placed in subsections.

Lines 420-422: please provide a reference or references.

Response: we have referenced the extensive review of Ho and colleagues on the topic of intrathecal amphotericin B treatment (Clin Infect Dis 2017).

Line 428: “null” should be “nil.”

Response: text was corrected.